# Dynamics of Acute Infection with *Mammarenavirus Wenzhouense* in *Rattus norvegicus*

**DOI:** 10.3390/v16091459

**Published:** 2024-09-13

**Authors:** Shanshan Du, Xuefei Deng, Xiaoxia Huang, Tiezhu Liu, Aqian Li, Qin Wang, Tingting Tian, Chuan Li, Zhangqi Zheng, Qihan Lin, Zhuowei Li, Shiwen Wang, Jiandong Li

**Affiliations:** 1National Key Laboratory of Intelligent Tracking and Forecasting for Infectious Diseases, National Institute for Viral Disease Control and Prevention, China CDC, Beijing 102206, China; duss@ivdc.chinacdc.cn (S.D.); dengxf456@163.com (X.D.); huangxx@ivdc.chinacdc.cn (X.H.); liutz@ivdc.chinacdc.cn (T.L.); liaq@ivdc.chinacdc.cn (A.L.); wangqin@ivdc.chinacdc.cn (Q.W.); tiantt@ivdc.chinacdc.cn (T.T.); lcehfcdc@163.com (C.L.); zairy_zzq@163.com (Z.Z.); linqihan12178@163.com (Q.L.); lizhuoweiccdc@163.com (Z.L.); wangsw@ivdc.chinacdc.cn (S.W.); 2NHC Key Laboratory of Biosafety, National Institute for Viral Disease Control and Prevention, China CDC, Beijing 102206, China; 3The Key Laboratory of Environmental Pollution Monitoring and Disease Control, Ministry of Education, School of Public Health, Guizhou Medical University, Guiyang 550025, China; 4NHC Key Laboratory of Medical Virology and Viral Diseases, National Institute for Viral Disease Control and Prevention, China CDC, Beijing 102206, China

**Keywords:** Wenzhou virus, acute infection, virus shedding, transmission, rodents, arenavirus

## Abstract

While *Mammarenavirus Wenzhouense* (WENV) is broadly distributed across Asia, the dynamics of WENV infection remain unclear. In this study, a field-derived strain of WENV was used to inoculate Sprague Dawley (SD) rats by intramuscular injection, and the process of viral infection was observed over the course of 28 d. Viral RNA became detectable in the blood at 3 dpi and remained detectable for about 12 d. In most organ tissues, viral RNA peaked at 7 dpi, and then began to decline by 14 d, but remained detectable in intestine and brain tissues at 21 and 28 dpi. Viral shedding was detected from fecal samples for 5 d, from 6 to 11 dpi using qRT-PCR, and was recovered from feces collected at 8 dpi. Horizontal contact infection occurred among cage-mates at 14 and 21 dpi. Antibodies against the nucleocapsid were detected at 5 dpi, and then increased and persisted until the end of the experiment. These results enabled us to determine the kinetics of viremic response, viral shedding in feces, and horizontal transmission dynamics, as well as the potential sites for WENV replication and viral maintenance in nature.

## 1. Introduction

*Mammarenavirus Wenzhouense* (WENV) is a single-stranded negative-sense RNA virus belonging to the family of *Arenaviridae* [1]. WENV was first identified in Asian house shrews in China and named after its place of discovery [2]. Subsequently, it was found that the virus was widely distributed across Asia and was identified in a wide range of host animals, such as *R. norvegicus*, *R. rattus*, *R. losea*, *R. exulans*, *R. tanezumi*, *R. nitidus*, *Niviventer niviventer*, *Suncus murinus*, *R. flavipectus*, *Mus musculus*, *Apodemus agrarius*, and *T. belangeri*. Viral RNA has been detected in multiple organs and tissues from field rodents, including the heart, liver, spleen, lungs, intestines, and the respiratory tract [3,4,5,6,7,8]. WEN-specific IgG antibodies have been detected using ELISA with a positive rate of 17.4% (89/510) in sera collected from patients with dengue/influenza-like illnesses [5,6], and a seroprevalence rate of 4.6% was reported in a serological survey in the general population [4]. Mammarenaviruses are generally associated with a specific rodent species as a reservoir, and can be asymptomatically shed via the feces, urine, and saliva of the infected rodent for weeks or months [9]. Humans and host animals usually become infected through contact with infected rodents or through the inhalation of infectious rodent excreta or secretions [10,11]. The pathogenicity of WENV in humans remains unclear; however, the evolution of WENV under changing natural selection pressures may make it a potential threat to human health. It has been accepted that pathogens within their hosts are not isolated but interact and evolve within a large community of microorganisms that can be commensal or pathogenic [12,13]. The co-circulation and co-infection of WENV with Hantaan or Seoul viruses in small mammals and humans has been recently reported [8], which could affect virus–rodent interactions and the eco-evolutionary and virological perspectives of the two viruses, while posing a risk, with the emergence of recombinant viral variants having increased fitness for circulation and zoonotic potential.

So far, there have been only a few reports on the dynamic characteristics of WENV infection in rodent hosts. In this study, we evaluated the process of WENV replication and shedding in infected SD rats by controlling time since inoculation, which may enhance our understanding of WENV transmission within reservoir populations and provide helpful information for the early warnings and predictions of emerging infectious diseases originating from rodents.

## 2. Materials and Methods

### 2.1. Animal Handling and Viral Inoculations

Animals were handled using approved protocols according to guidelines from the National Institute for Viral Disease Control and Prevention, China, CDC. At 6–8 weeks of age, 52 male SD rats (*Rattus norvegicus*) were inoculated intramuscularly with WENV (strain DG4) suspended in 0.2 mL of Eagle minimum essential medium (EMEM). WENV was isolated from the lung of a field rodent *Rattus flavipectus* and passaged one time in SD rats [8]. Homogenates intended for inoculation were diluted with a 10-fold excess of EMEM, and sham-inoculated control rats were inoculated intramuscularly with tissue homogenates from an uninfected rat. Two rats were raised in each individual ventilated cage (IVC). A total of 54 animals were divided into 4 groups; group 1 was set to include 5 IVCs, where both rats in each IVC were inoculated; group 2 included 20 IVCs, where 1 rat was inoculated with the virus and the other was sham-inoculated in each IVC, which was set to evaluate the possibility of WENV’s horizontal transmission to cage-mates in the IVC system; group 3 included 1 IVC, and both animals in the IVC were sham-inoculated as a control; group 4 included 2 rats in 1 IVC as a no-injection control. The vital signs of the rats, including weight, fur, trauma, emotional states, mucosal elasticity, and behavioral activity, were observed and recorded daily until the end of the experiment. Blood samples were collected from each rat from group 1 prior to infection, and then every other day post-inoculation via their tails using a capillary tube. Fecal samples were obtained from rats in each IVC from group 1 every day post-inoculation (pi). On days 7, 14, 21, and 28 pi, 5 cages of rats from group 2 were anesthetized with tribromoethanol, blood samples were collected from arterial vessels in the axilla at the side of the thorax, and multiple organs, including the heart, liver, kidneys, lungs, spleen, thymus, testes, brain, and intestines, were collected from the animals. After the samples were collected on day 28 pi, all animals were killed in the manner described above.

### 2.2. Enzyme-Linked Immunosorbent Assay

Plasma was used to detect anti-WENV immunoglobulin G (IgG) using an enzyme-linked immunosorbent assay; here, microtiter plates were coated overnight at 4 °C with a purified recombinant produced nucleocapsid (N) protein (0.4 μg/well) of WENV or recombinant N protein of Seoul virus diluted in a carbonate buffer, as described previously (8). Thawed plasma samples, as well as positive and negative control samples, were diluted 1:100 in phosphate-buffered saline (PBS)–Tween (PBS-T) with 5% skimmed milk powder and added in duplicate to antigen-coated wells. The plates were incubated at 37 °C for 1 h and washed with PBS-T, and secondary antibodies (horseradish peroxidase (HRP)-conjugated anti-rat IgG diluted 1:1000 or HRP-conjugated N protein in PBS with 5% skimmed milk powder) were added. The plates were incubated for 1 h at 37 °C and washed with PBS-T, and a TMB peroxidase substrate buffer (3, 3′, 5, 5′tetramethylbenzidine and hydrogen peroxide) was added to each well, which was terminated after 10–15 min by adding 2 N H2SO4 to each well. The optical density (OD) was measured at 450 nm with a reference wavelength of 620 nm, and the average OD for N protein of Seoul virus duplicates was set as negative control. Samples were considered positive if the OD value was ≥2.1 times of the values of the OD from negative control wells.

### 2.3. Quantitative TaqMan Reverse-Transcription PCR (qRT-PCR)

Aliquots of 100–200 mg of tissue or feces or 140 µL of blood from the SD rats were used to prepare RNA. The tissue and feces were placed in a 2 mL tube containing 0.5 mL Dulbecco’s modified eagle medium (DMEM) and mechanically homogenized using a Micro Smash machine MS-100 (TOMY; Tokyo, Japan). The homogenates were centrifuged at 12,000× *g* for 5 min, and the supernatant was preserved at −80 °C for RNA extraction and virus isolation. An RNeasy Mini Kit (Qiagen, Hilden, Germany) was used to extract the total RNA from 140 µL of the homogenates supernatant, and a QiaAmp viral RNA Mini Kit was used for RNA extraction from 140 µL blood samples, following the manufacturer’s instructions. A TaqMan quantitative real-time PCR (qRT-PCR) assay was performed, as previously described [8]. A one-step fluorescent quantitative RT-PCR kit (AgPath-ID Onestep RT-PCR Kit, ABI) was used for viral RNA detection. The reaction conditions were as follows: 50 °C for 30 min; 95 °C for 10 min; 95 °C for 15 s; 60 °C for 45 s. These cycles were performed 40 times. A standard curve containing dilutions ranging from 1 × 10^1^ copies to 1 × 10^7^ copies of the template was used on each 96-well plate, and viral RNA copy numbers were determined according to the standard curve; the cut-off cycle threshold (Ct) value for a positive reaction was set at 35 cycles (Figure 1).

### 2.4. RT-PCR and Sequencing

To examine whether the WENV strain had undergone mutations during the passage from the field-captured specimen and later passages in SD rats, we compared the complete sequence of the L segment and the S segment. These were obtained from the lungs from the field-captured *Rattus norvegicus* and those of the passage specimen; this was conducted through the direct sequencing of the viral amplification products. The genomic RNAs of the L and S segments were amplified through RT-PCR with a Taq PCR Master Mix Kit (Sangon, China), using degenerate primers. The PCRs were set up with a final volume of 50 µL. The thermal cycling conditions involved an initial denaturation period of 2 min at 95 °C, followed by 30 cycles of 20 s at 95 °C, 30 s at 55 °C, and 30 s at 72 °C. The final extension lasted 10 min at 72 °C. Each PCR product was sequenced by Sanger sequencing. The obtained viral genomic sequences and amino acid sequence of the open-reading frame (ORF) were aligned by Clustal W embedded in Mega11 software [14].

### 2.5. Virus Isolation

To evaluate WENV replication and shedding in SD rats, the homogenate supernatants of the qRT-PCR-positive lung and feces tissue samples were diluted in DMEM containing ampicillin (400 µg/mL) and streptomycin (100 µg/mL) with a ratio of 1:10. An additional gentamicin (50 µg/mL) was added in the dilutions of the homogenates of feces. A measure of 500 µL of each diluted solution was loaded onto freshly prepared Vero, with DH82 monolayer cells in T25 cell culture flasks, respectively, following common virus isolation technology using mammalian cells; measures of 200 µL of each of the dilutes were inoculated in SD rats with intraperitoneal injection (i.p.). The cells for virus rescue were incubated in 5 mL maintenance DMEM medium containing 2% FBS at 37 °C, supplemented with CO_2_ 5% for 10 days after the absorbance of the loaded homogenates dilutes, conducted through intermittently shaking the mixture for about 2 h. Then, the cells were passaged twice, and the results of the virus rescue procedure were determined using an immunofluorescence assay with polyclonal rabbit serum used against the recombinant N protein of WENV; the qRT-PCR method was employed.

### 2.6. Statistical Analyses

The copies of the detected viral RNAs were calculated using a standard curve containing dilutions of the in vitro-transcribed reference viral RNA. The statistical significance was analyzed using either the Pearson χ2 or continuity correction χ2 test. Analyses were performed using the SPSS software (ver. 16.0). Values of *p* < 0.05 were considered statistically significant.

## 3. Results

### 3.1. Virus Preparation

To prepare the WENV (strain DG4) stock, five 6–8-week-old SD male rats were inoculated with the homogenate of a WENV RNA-positive lung tissue from wild Rattus norvegicus captured in Jiangxi Province, China. The five rats received 200 μL of the homogenate at a 1:10 dilution in EMEM through intraperitoneal injection (I.P.). Blood samples were collected from the tail every other day after day 1, prior to inoculation, to examine them for viral RNA using qRT-PCR and the seroreactivity of the samples to the N antigen via ELISA. They were killed at 7 d, and a viral stock was prepared from a pooled homogenate of lung and spleen, which was used for all subsequent infections. There were about 5 × 10^7^ viral RNA copies of this homogenate/mL, as determined using qRT-PCR.

### 3.2. General Conditions of the Inoculated SD Rats

No significant symptoms were observed from the SD rats during the 4-week post-inoculation period with WENV stocks compared to the sham-inoculated ones. The animals were killed on 7, 14, 21, and 28 dpi, respectively. The heart, liver, kidney, lungs, spleen, thymus, testicles, brain, and intestine were collected. Bleeding points were only observed in the lungs and thymus of the inoculated rats; these were collected at 14 dpi and 21 dpi. No lesions were observed in any other collected organs.

### 3.3. Viral RNA and Plasma Antibody Responses in Tail Blood

Blood samples were collected from each rat from group 1 prior to infection and every other day post-inoculation from the tail using a capillary tube. Viral RNAs and viral-specific antibodies were tested using qRT-PCR and ELISAs. Viral RNA was detected at 3 dpi with a level of (1.67 ± 0.28) × 10^7^ copies/mL. It remained detectable for about 12 days until 15 dpi, with a level of (2.10 ± 0.28) × 10^5^ copies/mL; then, it peaked at 7 dpi, with the number of viral RNA being in the range of (9.52 ± 0.22) × 10^8^ copies/mL (Figure 2A). Anti-N antibodies were evident in tail blood by day 5 in three of the five samples; however, seroconversion was complete in all of the remaining animals at later time points. The OD values reached their maximum at 19 dpi and remained at this level in the later stages (Figure 2B).

### 3.4. Dynamic of Experimental WENV Infection in SD Rats 

Animals from group 2 were killed on 7, 14, 21, and 28 dpi, respectively, to examine five cages of SD rats in the IVC system at each interval using qRT-PCR and ELISA. Therefore, a minimum of five infected rats and five sham-inoculated rats were tested.

Of the five inoculated rats examined at 7 d, all five animals were viral RNA-positive in the heart, liver, kidney, lung, spleen, intestine, testes, and thymus gland; the number of viral RNA copies detected in the lung, thymus, and spleen, at a range of approximately 5.60~8.81 × 10^6^ copies/g, was significantly higher than that in other tissue samples (*p* < 0.01) (Figure 2). By day 14, except for the brain and testes, the other tissues of the five animals were positive for viral RNA, although the RNA load decreased (Figure 3), which coincided with the appearance of high titers of antibodies in the blood. By 21 d and onward, the five animals examined showed viral RNA in the tissue samples with a significantly decreased number of copies; this is with the exception of the liver, kidney, thymus gland, and testis, where it was undetectable in all specimens (Figure 3). In brain tissue, viral RNA was only detected in samples collected on days 21 and 28 pi (Figure 3).

For the five sham-inoculated rats examined at each interval, at the same time as that for the inoculated rats in the same cages, WENV RNA was detected in the lung tissue samples from two out of five rats at 14 dpi and in the lung tissue samples from one out of five rats at 21 dpi using qRT-PCR; these data suggest that horizontal contact infection occurred among cage-mates.

Arterial blood samples were also collected on 7, 14, 21, and 28 dpi. Viral RNA was detected in the arterial blood using qRT-PCR from all the animals at 7 d and 14 d, with a mean level of 1.82 × 10^7^ copies/mL at day 7; however, at the other time points, no animal tested positive for the viral RNA (Figure 4A). Anti-N antibodies were detectable in all five inoculated rats collected at 7 dpi, and they reached their maximum at 21 dpi (Figure 4B).

### 3.5. Virus Shedding and Rescue

Fecal samples were collected from cages with inoculated SD rats of group 1 everyday post-inoculation. WENV RNA in feces was detected on 6 dpi and remained detectable at a range of 10^4^ copies/g for 5 days until 11 dpi using qRT-PCR; meanwhile, it could not be detected at other time points (Figure 5A). The PCR amplification products obtained from feces were sequenced and verified as WENV RNA. To evaluate the infectivity of WENV shedding in the feces of inoculated SD rats, the freshly prepared homogenates of the feces samples were inoculated onto monolayer Vero and DH82 cells; this procedure was also performed in SD rats with intraperitoneal injection (i.p.). The development of viral RNA was detected in the tail blood from the inoculated SD rats with freshly prepared homogenates of feces, with a level of (9.55 ± 0.31) × 10^7^ copies/mL WENV RNA at 7 dpi (Figure 5B). Additionally, antibodies against N protein were detected after 9 dpi in the blood collected from the tail (Figure 5C). However, the procedure used in this study failed to recover WENV from the fecal samples through a mammalian cell culture, as has been reported in previous studies [7,8]. The authors of this study sought to determine whether the experimental passage of WENV through rats may have selected for a genetic variant that is more readily transmitted under experimental conditions than the original wild virus; in order to do this, the complete sequences of the L and S genomes from the wild-caught rats and the lung samples from the rats of passages 2 and 3 were collected at 7 dpi to be sequenced and compared. However, no variants were detected.

## 4. Discussion

Rodents represent the largest order of living mammals with wide distribution, diverse species, strong reproductive ability, and strong adaptability in the world [15]; they are known to be reservoir hosts for at least 60 zoonotic diseases [16]. WENV was first identified in Wenzhou China, and was subsequently found in broad areas of China, Cambodia, Thailand, and Laos, demonstrating the wide distribution of WENV infection in rodents and humans in Asia [3,4,5,6,7,8]. The rodent hosts of WENV, such as *Rattus norvegicus* and *Mus musculus*, are commonly found in human dwellings and in peridomestic habitats, which could facilitate the frequent spillover of the virus to humans [17].

In this study, we intended to reveal the dynamics of WENV infection in its rodent reservoirs at the acute phase. All animals remained asymptomatic within 28 days after inoculation of 6–8-week-old SD rats. The pathogenesis of arenavirus diseases is believed to involve initial replication at the site of infection in non-reservoir hosts, usually following aerosol deposition in the lung [18]. WENV replicated to high titers in the lung, thymus, and spleen of the inoculated rats (>10^6^ RNA copies per g) at 7 dpi; a low titer of viral RNA was detected in the heart, liver, kidney, brain, intestine, and testis (~10^5^ RNA copies per g). Lungs were found with the highest level of viral loads, which could be the primary site of replication for WENV. It was suggested that the viral replication seems to occur systematically throughout the acute phase of infection [19]. In brain tissue, viral RNA could be detected in samples at 21 and 28 dpi; meanwhile, in other organs, viral RNA could not be detected, except for in the liver, spleen, and small intestine. The results demonstrated that WENV could naturally infect brain tissue, with a delay of about 2 weeks compared to other organs, indicating that the virus could cross the blood–brain barrier and pose a potential risk of developing central nervous system diseases in humans after becoming infected. It was reported that the mechanism of arenavirus clearance from the organs of acutely infected rats was associated with a T-cell-mediated immune response to viral infection; this had been found in the clearance of lymphocytic choriomeningitis virus (LCMV) in infected animals [20]. Generally, it is accepted that the kinetics of virus clearance in different organs varies significantly; the virus is usually cleared from the liver within 30 days, but neurons can contain a viral antigen for 90 days or more, and the clearance of the virus and antigens from the kidneys requires more than 200 days [18]. A relative stable level of viral RNA was detected in the small intestine and brain tissues at 21 and 28 dpi in this study; whether or not that is related to the persistent infection of WENV remains unclear. Several studies carried out on hantaviruses highlighted that the transition from the acute phase to the chronic phase could occur at different times from the first 2–3 weeks to the first 2–3 months of infection [21,22,23].

In this study, a transient viral RNAemia of WENV occurred between 5 and 15 days in SD rats; this finding is similar to that presented by a previous report, where viral RNA was detected in sera at 3 dpi and remained detectable till 15 dpi [7]. The RNAemia detected remained at a low and stable level, and further research is needed to determine whether this is related to the tissue tropism of WENV and its impact on virus shedding from infected host animals. The isolation of WENV using mammalian cells such as Vero, BHK, and DH82 was unsuccessful in this study. The determinants of the period of viremia in the mature host are unknown; both the host genetic component and the viral genes play important roles in the maintenance of the persistent viremic response [18]. It is well known that the arenavirus readily infects mammals; however, strains of animals vary in their susceptibility to virus infection. For example, the MHA strain of hamsters is highly susceptible to Pichinde virus infection, while those of the LVG strain are resistant to this infection [18].

Only fecal samples were collected in this study to examine the shedding of WENV. Viral RNA was detected at 6 dpi, and transmission of WENV among cage-mates were detected at 14 (2/5) and 21 dpi. (1/5) under the IVC system. During the feeding period of the experimental animals, no significant mutual biting was observed between rats, and no obvious bite marks were found from the animals; this could indicate that the transmission is unlikely to have been caused by mutual biting between animals. Virus shedding could be affected by the mode of transmission, the quantity of virus inoculated, or some individual factors [19]. Arenavirus had been found in a various array of excreta/secreta of their infected rodent reservoirs, such as the urine, saliva, or feces [18]. WENV genomic RNA was detected from the respiratory tract samples of rodents collected from Cambodia, Thailand, and Laos [6]; in addition, periodic shedding could occur during the persistent phase of the arenavirus infection.

An important issue that should be experimentally addressed is that surrounding the kinetics of the appearance of humoral antibodies in the WENV infection; this will be helpful in understanding the immune mechanisms that occur under reservoir–WENV interactions. Antibody seroconversion was detected at 5 dpi, high levels of antibodies were found at 15 dpi, and the antibody response was found to increase and persist until the end of the experiment. During the process of LCMV infection in mice, a vigorous antibody response to LCMV antigens has been found to be insufficient in clearing the virus. However, antibodies which are complexed with circulating viral antigens become lodged in the renal glomeruli and arterial walls; this could lead to chronic glomerulonephritis and arteritis [18].

In summary, in this experimental infection of WENV in SD rats, by controlling time since inoculation, samples from several organs, the blood, and feces of the rats were analyzed simultaneously. This process enabled the authors of this paper to determine the kinetics of the viremic response, observe the process of virus shedding in the feces, monitor the occurrence of horizontal transmission, identify sites for viral replication at the acute phase of infection, and identify potential sites for viral maintenance. Although the underlying mechanisms of these processes warrant further investigation, the results provide important insights into the infection, transmission, and prevalence of WENV.

## Figures and Tables

**Figure 1 viruses-16-01459-f001:**
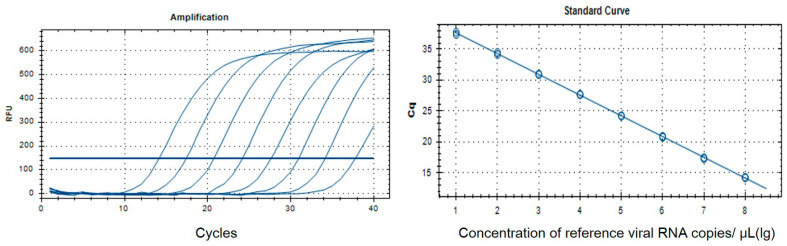
Amplification plots and standard curves of one-step real-time quantitative TaqMan RT-PCR assay for detection of WENV. The qRT-PCR assays were tested using synthesized in vitro target viral RNA transcripts ranging from 10^1^ to 10^8^ copies/mL. A PCR baseline subtractive curve fit view of the data is shown, with relative fluorescence units (RFUs) plotted against cycle numbers. Standard curves were generated from the Ct values obtained against known concentrations; the coefficient of determination (R^2^) and slope of the regression curve for each assay are Y = −3.364x + 41.01, R^2^ = 1.000.

**Figure 2 viruses-16-01459-f002:**
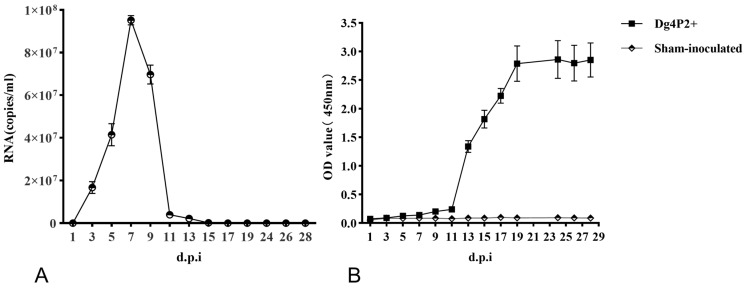
Development of viral RNA and anti-N antibodies in tail blood samples. Blood samples were collected from WENV-inoculated SD rats every other day and were examined for viral RNA using qRT-PCR (**A**) and reactivity to the WENV N antigen through ELISA (**B**). The RNA copy number is expressed as the mean of all blood samples with a detectable signal at each time point. The concentration of the antibodies is expressed as the mean of the optical density values obtained from all the blood samples. The error bars indicate the SEM for five blood samples.

**Figure 3 viruses-16-01459-f003:**
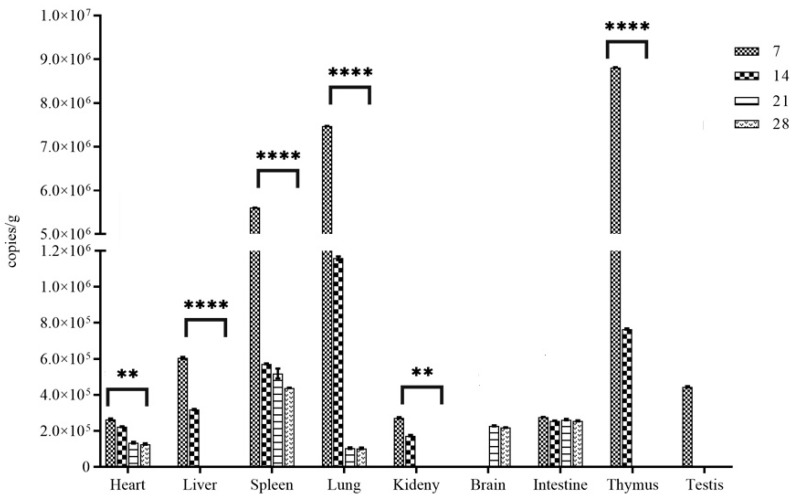
Quantitative TaqMan RT-PCR analysis of WENV RNA in tissues of inoculated SD rats. The RNA copy number is expressed as the mean of all the tissues with a detectable signal at each time point. Error bars indicate the SEM for three replicates. ** indicates the values of *p* < 0.05; **** indicates the values of *p* < 0.01.

**Figure 4 viruses-16-01459-f004:**
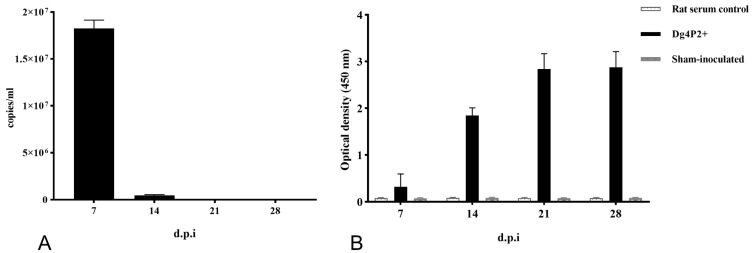
Development of viral RNA and anti-N antibodies in arterial blood. Sera from all WENV-inoculated SD rats at 7, 14, 21, and 28 dpi were examined for viral RNA using qRT-PCR; (**A**) reactivity to the WENV N antigen was examined using ELISA (**B**). The RNA copy number is expressed as the mean of all blood samples with detectable signal at each time point. The concentration of antibodies is expressed as the mean of optical density values obtained from all the blood. The error bars indicate the SEM for three replicates.

**Figure 5 viruses-16-01459-f005:**
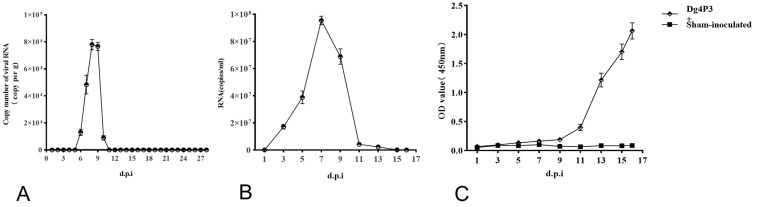
Virus shedding in feces from inoculated SD rats. Fecal samples were collected from cages with inoculated SD rats of group 1 everyday post-inoculation and were examined for viral RNA using qRT-PCR (**A**); blood samples were collected from the tail every other day and examined for viral RNA using qRT-PCR (**B**); antibodies against the WENV N antigen were detected using ELISA (**C**). The RNA copy number is expressed as the mean of all the samples with a detectable signal at each time point. The concentration of the antibodies is expressed as the mean of the optical density values obtained from the tail blood samples. The error bars indicate the SEM for five blood samples.

## Data Availability

The data underlying this article are available in this manuscript.

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
