# Peer review of "Dynamics of Acute Infection with Mammarenavirus Wenzhouense in Rattus norvegicus"

_viruses, 2024, doi:10.3390/v16091459_

Round 1

Reviewer 1 Report

Comments and Suggestions for Authors

I have no suggestions for the authors; I found the manuscript complete and well done. I would just suggest a language check.

Comments on the Quality of English Language

I have no suggestions for the authors; I found the manuscript complete and well done. I would just suggest a language check.

Author Response

Comments: I have no suggestions for the authors; I found the manuscript complete and well done. I would just suggest a language check.

We appreciate very much for the positive comments, we have checked the language of the MS.

Reviewer 2 Report

Comments and Suggestions for Authors

The authors put together a basic animal experiment regarding Wenzhouense virus and it’s behavior in rats. The manuscript is decently written but requires a moderate effort to refine the English. The format of the paper is acceptable. The content of the paper could use some buttressing with respect to the methods; in several places it is unclear what the authors did to achieve their claims. This manuscript requires a significant re-write in terms of the results, matching text to figures, as well as a significant refocusing on the claims; specifically, there is no evidence, in this reviewer’s opinion, that infections were achieved.

Why not perform a plaque assay on the viral inoculum to get a dose? This assay is the gold standard in addressing virus infectious doses. Stamping the inoculum with genomic copies per milliliter is helpful but the methods are not sufficient enough for another laboratory to determine exactly how the authors accomplished that quantification. It is unclear if copy number would be equivalent to genome number because the methods are not described. Is the standard curve used to construct this assessment of “copies” based on synthetic RNA? What was the source of the curve? Why does the curve not go above the copy number used to inoculate the animals?

The maximum copy number found in the blood does not reach 1 x 105, however the authors claim that the number was 105. Exact numbers should be used in the results, including averages and error bars. The authors also claim that the RNA was detectible in the blood to 15 days post-infection but the graph represents a bottoming out. This low level on the graph could be the function of the axis being inappropriately labeled but it could also be that this represents a limit of detection; the reader is unsure because the limits of detection are not discussed. Additionally, the amount put in exceeds the amount detected by several orders of magnitude; could it be that the animals were just clearing what was given without establishing an infection? The authors should redesign their figures to account for limit of detection and discuss the results in a way that considers the points raised above.

Figure 2 discusses RNA in copies per gram. In figure 1 the copies are discussed in terms of copies per microliter. Challenge titers are only discussed as “about 5 x 107 copies/mL.” The authors need to rewrite their text to either standardize their titers or more clearly state what volume and titer went where.

Figure 3 shows viral RNA in blood and antibodies for N. Again, the amount of RNA is several orders of magnitude less than the inoculum and only present on day 7. The text says that blood copies reached 107 but the figure does not support that number. The lack of antibodies from the control group would suggest that none were infected however the authors claim infection. Text and figures need to be realigned to agree with one another.

Figure 4 shows viral “presence” in the feces. Again, the total amounts are orders of magnitude off from the inoculum and claimed as 105 but never make it to or above 105 in the figure.

Discussion line 280 claims “transient viremia,” but this reviewer would disagree. What the authors have witnessed is transient RNA-emia. They found genetic material but there is no evidence in the manuscript that what was used as inoculum contained infectious virus. It’s completely possible that they used animals that had genetic material and protein, but the version of inoculum did not have infectious virus. The authors should have performed a plaque assay at some point to demonstrate that there was indeed infectious virus present. None of the figures/data support infection of the animals. Each figure shows that the measured amount of genetic material measured is orders of magnitude less than what was used in the inoculum and the timepoints in which they find that material is consistent with animals clearing a foreign substance, completely matching the organs surveyed as well. In fact, none of the uninfected animals demonstrated antibody titers in figure 3, further supporting live infection was not observed.

The whole manuscript requires re-writing and further analysis, most specifically for infectious virus.

Comments on the Quality of English Language

Just needs a once over to polish. 

Author Response

Comments 1: The authors put together a basic animal experiment regarding Wenzhouense virus and it’s behavior in rats. The manuscript is decently written but requires a moderate effort to refine the English. The format of the paper is acceptable. The content of the paper could use some buttressing with respect to the methods; in several places it is unclear what the authors did to achieve their claims. This manuscript requires a significant re-write in terms of the results, matching text to figures, as well as a significant refocusing on the claims; specifically, there is no evidence, in this reviewer’s opinion, that infections were achieved.

Response 1: We would like to express our sincere gratitude to the reviewers for their scientific and meticulous review of our manuscript. We have addressed the relevant questions as follows and made corresponding revisions in the manuscript.

Comments 2: Why not perform a plaque assay on the viral inoculum to get a dose? This assay is the gold standard in addressing virus infectious doses. Stamping the inoculum with genomic copies per milliliter is helpful but the methods are not sufficient enough for another laboratory to determine exactly how the authors accomplished that quantification.

Response 2: As stated in the results section of our manuscript, we were unable to isolate the virus through mammalian cell culture and only passaged it through intraperitoneal or intramuscular inoculation in animals. Similar reports have also been made by other researchers. This makes quantitative analysis of viruses through plaque titration difficult to overcome, as well as evaluating the immune response of experimental animals to infection through plaque reduction neutralization tests.

Comments 3: It is unclear if copy number would be equivalent to genome number because the methods are not described.

Response 3: We do not believe that the RNA copy number detected by qRT PCR calculated through standard curves will be exactly the same as the number of infectious virus particles or genomes, but can reflect the amount of virus, which should be the second best choice.

Comments 4: Is the standard curve used to construct this assessment of “copies” based on synthetic RNA? What was the source of the curve?

Response 4: Yes, the standard curve used in this MS based on synthetic RNA. We have added the amplification plots and standard curves of one-step real-time TaqMan RT-PCR assays as an Appendix Figure. The one-step realtime qRT-PCR assays were tested using synthesized in vitro target viral RNA transcripts ranging from 101 to 108 copies/mL. A PCR baseline subtractive curve fit view of the data is shown with relative fluorescence units (RFUs) plotted against cycle numbers. Standard curves generated from the Ct values obtained against known concentrations, the coefficient of determination (R2) and slope of the regression curve for each assay are indicated in the figure.

Comments 5: Why does the curve not go above the copy number used to inoculate the animals?

Response 5: The amplification curve of the WENV in the experimental rats did go above the copy number used to inoculate the animals. 0.2 ml of viral stock (homogenates) with a concentration about 5 × 107 copies/mL viral RNA copies determined by qRT-PCR, while a number of viral RNA copies at a range of 105 copies/μL was yielded in the blood samples. In the MS, we did not use a unified unit of measurement, which caused some confusion. We standardized the corresponding charts and descriptions to be in milliliters.

Comments 6: The maximum copy number found in the blood does not reach 1 x 105, however the authors claim that the number was 105. Exact numbers should be used in the results, including averages and error bars.

Response 6: Thanks a lot for the suggestion. We have added the exact numbers in the section of result.

Comments 7: The authors also claim that the RNA was detectible in the blood to 15 days post-infection but the graph represents a bottoming out. This low level on the graph could be the function of the axis being inappropriately labeled but it could also be that this represents a limit of detection; the reader is unsure because the limits of detection are not discussed. Additionally, the amount put in exceeds the amount detected by several orders of magnitude; could it be that the animals were just clearing what was given without establishing an infection? The authors should redesign their figures to account for limit of detection and discuss the results in a way that considers the points raised above.

Response 7: Thanks a lots for the kind suggestions. The RNA copy number on the 15th day was indeed relatively low, close to the cut-off value. We redesigned the figures and added specific numerical descriptions. And we believe that the WENV was replicated in the inoculated rats as described above, and have standardized the unit of measurement to clear the confusion.

Comments 8: Figure 2 discusses RNA in copies per gram. In figure 1 the copies are discussed in terms of copies per microliter. Challenge titers are only discussed as “about 5 x 107 copies/mL.” The authors need to rewrite their text to either standardize their titers or more clearly state what volume and titer went where.

Response 8: In order to make it easier to calculate the amount of virus when inoculating experimental rats, we calculated concentration of the virus storage solution in a format of copies/mL, while to discuss virus in the tissues, RNA copies per gram was used, which should make it easier to compare results between laboratories.

Comments 9: Figure 3 shows viral RNA in blood and antibodies for N. Again, the amount of RNA is several orders of magnitude less than the inoculum and only present on day 7. The text says that blood copies reached 107 but the figure does not support that number. The lack of antibodies from the control group would suggest that none were infected however the authors claim infection. Text and figures need to be realigned to agree with one another.

Response 9: As discussed at A5 and A7, WENV infection in inoculated rats was proved in this MS, N protein specific antibodies were detected in the blood of the inoculated rats but not in the blood collected from negative control and sham-inoculated animals.

Comments 10: Figure 4 shows viral “presence” in the feces. Again, the total amounts are orders of magnitude off from the inoculum and claimed as 105 but never make it to or above 105 in the figure.

Response 10: The levels of detected WENV shedding in the feces were not as high as detected in the lung tissues, and the exact number of RNA copies of the rescued WENV in the blood was presented in the text. 

Comments 11: Discussion line 280 claims “transient viremia,” but this reviewer would disagree. What the authors have witnessed is transient RNAemia. They found genetic material but there is no evidence in the manuscript that what was used as inoculum contained infectious virus. It’s completely possible that they used animals that had genetic material and protein, but the version of inoculum did not have infectious virus. The authors should have performed a plaque assay at some point to demonstrate that there was indeed infectious virus present. None of the figures/data support infection of the animals. Each figure shows that the measured amount of genetic material measured is orders of magnitude less than what was used in the inoculum and the time points in which they find that material is consistent with animals clearing a foreign substance, completely matching the organs surveyed as well. In fact, none of the uninfected animals demonstrated antibody titers in figure 3, further supporting live infection was not observed. The whole manuscript requires re-writing and further analysis, most specifically for infectious virus.

Response 11: Thanks a lot for the suggestions. We agree that “transient viral RNAemia” is more accurate than “transient viremia”, and has been revised the claims accordingly. As stated at A2, we did not successfully isolate WENV through mammalian cell culture as some other researchers, which makes it to quantify the virus using the golden standards of plaque titration, but the definitely increasing of virus replication the experimental animal’s multi organ tissues and blood, and the sero-conversion and next increasing of specific antibodies in the blood showed robust evidences of successful infection of WENV in the experimental rats. The RNAemia detected in this study remained at a low and stable level, and further research is needed to determine whether this is related to the tissue tropism of WENV and its impact on virus shedding from infected host animals.